# Municipality Data as a Rapid and Effective Tool to Analyse Spatial and Temporal Variations of All-Cause Mortality by Town District: The Experience in Genoa (Italy)

**DOI:** 10.3390/ijerph18168250

**Published:** 2021-08-04

**Authors:** Paolo Contiero, Giovanna Tagliabue, Andrea Tittarelli, Martina Bertoldi, Claudio Tresoldi, Giulio Barigelletti, Viviana Perotti, Vittoria Balbo, Stefania Rizzieri, Marco D’Orazi, Valerio Gennaro

**Affiliations:** 1Environmental Epidemiology Unit, Fondazione IRCCS Istituto Nazionale dei Tumori, 20133 Milan, Italy; paolo.contiero@istitutotumori.mi.it (P.C.); martina.bertoldi@istitutotumori.mi.it (M.B.); 2Cancer Registry Unit, Fondazione IRCCS Istituto Nazionale dei Tumori, 20133 Milan, Italy; claudio.tresoldi@istitutotumori.mi.it (C.T.); giulio.barigelletti@istitutotumori.mi.it (G.B.); viviana.perotti@istitutotumori.mi.it (V.P.); 3Information Office of the Municipality of Genoa, 16149 Genoa, Italy; vbalbo@comune.genova.it (V.B.); srizzieri@comune.genova.it (S.R.); mdorazi@comune.genova.it (M.D.); 4International Society of Doctors for the Environment (ISDE), Past Director of the Liguria Mesothelioma Registry, Ospedale Policlinico S. Martino, 16132 Genoa, Italy; valerio.gennaro52@gmail.com

**Keywords:** environmental health inequalities, environmental health justice, urban health, mortality analysis

## Abstract

The main objective of this study was to analyse the space–time epidemiological differences by sex during the 2009–2020 period in the total mortality recorded among residents in each of the 25 districts of the Genoa municipality, net of the age effect. The analysis was based on official statistical data relating to total mortality and on the resident population. An estimate of the expected deaths was made to calculate the sex-specific age-standardised mortality ratio (SMR). The temporal trends and age-standardized death rates (SDRs) with respect to those of the European population specific to sex and calendar year were identified for each district. Over the entire observation period, the SMR for males ranged from 124.4 (Cornigliano) to 82.0 (Albaro); for females, the values ranged between 133.4 (Cornigliano) and 85.6 (Nervi-Quinto-S. Ilario). Between 2019 and 2020, Genoa recorded an increase in SDR of 24.5%, more pronounced in males (+26.7%) than in females (+22.4%). This epidemiological methodology is replicable and allows to quickly identify spatial, temporal, sex, and age differences in the general mortality within a municipality.

## 1. Introduction

In 1452, the Duke of Milan Francesco Sforza established the *Mortuorum Libri* of Milan (1452–1801) [1]. The *Mortuorum Libri*, or Death Registers, were based on the daily records of deaths in the city and were conceived as a tool for the swift containment of contagious diseases with rapid interventions such as closure of the affected homes or transfer of the sick to the lazaretto. Because of this strong public-health objective, the parishes where the deceased individuals lived or where the deaths occurred were systematically reported. In England, a mortality registry was set up during the plague outbreak in 1532; the *Bill of Mortality* [2] reported the weekly mortality data in the city of London by parish. These two examples highlight the importance of a rapid-information health system; the Milan registry made the number of deaths available day by day and the London one week by week, in both cases with geographic detail by parish so that the necessary interventions could be planned. Similar information systems rapidly describing the mortality by neighbourhood or district are not routinely available in our time; more in general, mortality data are not available with the same speed as in the Milan and London examples. However, some other experiences have been described: a study in New York City highlighted an excess mortality of 50% in the Central Harlem health district, and another important study performed in Barcelona revealed marked differences between the mortality rates of the 38 neighbourhoods of the city [3].

Based on these premises, we hypothesised that the death counts as provided by municipal statistics offices could be a suitable source of information for the rapid and geographically detailed analysis of relevant within-town mortality differences [4]. For our study, we used the mortality counts registered by the municipal statistics office of Genoa, a coastal city in the north-west of Italy with the largest port in the country and one of the most important ones in Europe, harbouring many industrial activities. The aims of the study were as follows: (a) to test the possibility of producing general mortality statistics with a delay of a few months from the events; (b) to test if the situation at the district level could provide useful insights into population health; and (c) in the case of perceived differences in mortality by district, to test if these differences persisted or diminished over time.

## 2. Materials and Methods

### 2.1. Population under Study

The average annual population of the city of Genoa in the 2009–2020 period was approximately 591,000 people, including 278,000 males and 313,000 females. The surface of the city is 240.29 km^2^, with a population density of 2484.3 inhabitants/km^2^.

The distribution of the Genoa population according to sex and age class (by 1 January 2021) is shown in Appendix A.

Using data of the Italian National Institute of Statistics (ISTAT) [5] to analyse the age structure of the population of Genoa considering three age groups (young, <15 years; adult, 15–64 years; and older, >64 years), we point out that it is regressive, with the proportion of young people being smaller than that of the older population. This phenomenon is even more marked than it is in the Italian population overall (see Appendix A). Some indicators (shown in Appendix A) could be useful to evaluate the social implications of this phenomenon more in detail [6].

### 2.2. Study Design and Statistical Analysis

This study was designed as an ecological investigation based on the death counts of the resident population as registered by the municipal statistics office for the 25 districts (neighbourhoods) comprising the municipality, and the mortality rates in Genoa over the years 2009–2020. The death counts were registered without a cause of death, as is usual for municipality data.

Using the mortality data of the city of Genoa as a whole (8306 deaths/year on average, 3825 males and 4481 females), we estimated the expected deaths for each district to calculate the age-standardised mortality ratio (SMR) and the 90% confidence limits by year and specific to sex.

As the resident population was rather small in some districts, we decided to calculate SMRs for the entire time period, that is, 2009–2020, and to use 90% confidence intervals (90% CI), which are considered better suited to provide results that underpin the principles of precaution as well as prevention [7].

The temporal trend and the age-standardised death rate (SDR) with respect to those of the European population, specific to sex and calendar year, were identified for each district.

Finally, to obtain further information on the temporal trends for each year and both sexes, the differences between the highest and lowest SDRs in the various districts were calculated.

Figure 1 shows the Genoa municipality divided into 25 districts, while Appendix A reports the surfaces of each district.

#### 2.2.1. SMR

The indirect method (by calculating the SMR) allows to analyse mortality figures among populations with different age structures. The SMR is the ratio between the number of observed deaths (*OBS*) in the study population and the expected deaths (*EXP*) if the mortality of a standard population is taken as the reference (in our case, that of the municipality of Genoa). It is usually multiplied per 100 so that it can be expressed as a percentage.

The expected deaths of each district *i*, for the whole period of twelve years (2009–2020), were thus calculated by applying the corresponding specific mortality rates (*MR_i_*) of the municipal population to the total population (*N_i_*) by age and sex of each district of the municipality of Genoa.

The SMR in each district for the whole period was calculated as the sex-specific sum of observed deaths (numerator) divided by the number of deaths expected therein (denominator).

Equation (1) was used to calculate the SMR for sex:(1)$$SMR=\frac{OBS_{i}}{EXP_{i}}\times 100=\frac{OBS_{i}}{\sum^{\;}MR_{i}\times N_{i}}\times \;100$$
where *OBS_i_* is the number of deaths observed in district *i*; *EXP_i_* is the number of expected deaths in district *i*; *MR_i_* is the mortality rate of the municipal population in the *i*-th age group; and *N_i_* is the total population of district *i*.

To estimate the SMRs, the R function ageadjust.indirect (Package epitools version 0.5–10.1), which calculates age-standardised (adjusted) rates and exact confidence intervals, was used [8,9].

#### 2.2.2. SDR

For each district and for the whole municipality of Genoa, the SDRs per 10,000 inhabitants by year and sex were calculated applying the direct method [10] (R function ageadjust.direct, Package epitools version 0.5–10.1). The European Standard Population (ESP 2013) with age distribution was used as the reference population [11].

The mortality rates by age group, sex, and year for each district and for the municipality of Genoa as a whole were calculated by dividing the number of deaths observed in each age group by the corresponding number of the average annual population of each district and the municipality of Genoa.

The SDR of each district and of the municipality was thus obtained by the sum of the number of deaths resulting from applying the specific mortality rates to the corresponding number of the age group of the ESP used as the reference.

## 3. Results

### 3.1. SMR Analysis

Figure 2 shows the SMRs (with 90% CIs) by sex calculated over the whole period (2009–2020) for each of the 25 districts of Genoa, ordered by decreasing values in males. The highest SMRs among males were found in the districts of Cornigliano (124.4) and Pra’ (124.0), while the lowest was observed in the district of S. Francesco d’Albaro (82.0). For females, the district of Cornigliano presented a very high SMR (133.4), while three districts had values below 90: Valle Sturla (87.1), S. Francesco d’Albaro (86.5), and Nervi-Quinto-S. Ilario (82.0). Eight districts for males and five districts for females had statistically non-significant SMRs at the 90% level, with the interval crossing the reference value of 100. These SMRs are shown in Appendix A for males and females, respectively.

### 3.2. SDR Analysis

To show an example of the differences in SDRs by district, in Figure 3 and Figure 4, the annual SDRs calculated for the districts of Pra’ and Pegli, respectively, both compared to the annual SDRs of the whole municipality, are presented divided by sex. The two districts are located in the same area (West Genoa), but they show important differences in terms of health status of the population.

In Appendix A, the annual SDRs for each district and the municipality of Genoa as a whole are presented for males and females, respectively.

The annual SDR trends allowed to highlight important temporal variations within specific districts (data not shown) and throughout the city. In particular, between 2019 and 2020, Genoa saw an increase in SDR of 24.5%, which was more pronounced in males (+26.7%) than in females (+22.4%). The increase comprised almost all districts and was greatest in S. Teodoro (+50.7%), most markedly among females (+67.2%; males +42.3%). The only exception was the district of Foce (−6.6%). In Figure 5, the percentage differences between 2020 and 2019 divided by sex are shown for each district and for the municipality of Genoa as a whole. Some districts had a decreased SDR: Foce and Struppa among males; and Foce, S. Francesco d’Albaro, Portoria, and Pre-Molo-Maddalena among females. The differences in SDRs between 2019 and 2020 were probably due to the SARS-CoV-2 pandemic.

To describe the geographical variability by district, a useful indicator is the difference between the highest and lowest values of the SDRs in the various districts. These were calculated for each year and are presented in Figure 6, divided by sex, where diff % SDR M represents the percentage difference between 2020 and 2019 in males, while diff % SDR F represents the percentage difference between 2020 and 2019 in females. The differences tended to increase slightly during the observation period, with a strong increase in 2018 compared with 2017, followed by a marked decrease in 2019, and again a small increase in 2020. To interpret the variation of SDR within the city of Genoa, we took as reference the variation of SDR between the 20 regions of Italy and the variation of SDR between the 111 Italian provinces [12]. In 2016, the region with the lowest SDR for males was Trentino (94.27), while the one with the highest SDR was Campania (117.47), 24.6% more than Trentino. For females, the region with the lowest SDR was again Trentino (61.55), while the highest SDR was recorded in Campania (80.19), 30.3% more than Trentino. In 2016, the province with the lowest SDR for males was Monza Brianza (90.55), while the highest SDR was found in Caserta (124.3), 37.2% more than Monza Brianza. For females, the province with the lowest SDR was Pordenone (61.55) and the one with the highest SDR was Napoli (85.29), 38.5% more than Pordenone. The average SDR for Italy was 101.99 for males and 68.61 for females.

To test the comparability of the data, we also performed a comparison between the mortality figures from the municipal statistics office of Genoa and those registered by the Italian Statistical Institute (ISTAT) [13]. In Appendix A, the figures from these two sources are presented for the last six years considered in our study (2015–2020), showing that ISTAT counted a slightly higher number of total deaths than those supplied by the municipality.

## 4. Discussion

The first aim of the work described in this paper was to understand whether it is possible to rapidly produce mortality data by city district. The answer to this question was affirmative: mortality statistics as the one we showed in this paper could be easily produced in few months and, in addition, it is possible to monitor any increase in mortality counts nearly day by day, as in the historical example of the outbreak in Milan and London. The answer to the second question, whether the mortality data could reveal important differences between districts with implications for population health, was also positive: some districts showed an excess mortality (difference between observed and expected deaths) of 24%, while others displayed more than 16% under the Genoa average. The third question posed in this study was whether or not the differences between districts were attenuated over time; in this case, we observed that the existing differences persisted.

The need for timeliness is unfortunately much greater in the era of COVID-19 [14], where any delay in the detection of a threat and, therefore, in the planning of emergency measures means we will be faced with many victims. The example of COVID-19, or other epidemics such as the plague, is also useful to explain the need for data on a sufficiently detailed geographic scale. In the case of the *Mortuorum Libri* of Milan, the availability of daily mortality counts by parish was useful to attempt rapid containment of the spread of contagious diseases with immediate measures such as closure of the affected homes or transfer of the sick to the lazaretto. 

The analysis of SDR differences by city districts or neighbourhoods is not common in the scientific literature, although some experiences have been described. One of the most important ones is that of Barcelona [15,16], where an analysis of mortality performed in an urban setting revealed marked differences between the SDRs of the 38 neighbourhoods of the city, ranging from 84.47 to 152.99 in males (+81%) and from 85.22 to 157.15 in females (+84.4%). 

A very interesting paper was published about the excess mortality in the Central Harlem health district of New York City [3]. It reported that the age-adjusted rate of mortality from all causes in that district was the highest in New York City, more than twice that of U.S. whites and 50% higher than that of U.S. blacks. Another 54 health districts (of 353) in New York City showed age-adjusted mortality rates more than twice as high as those expected based on the U.S. white age-adjusted death rates. Another study analysing mortality inequalities by neighbourhood in 1989–1991 versus 1999–2001 [17] highlighted that, in the latter period, SDRs in the poorest communities were 50% higher than those in the wealthiest ones, and that relative inequalities only slightly decreased between the two periods.

Interestingly, the differences we identified in Genoa between districts were markedly greater than those observed between the different regions and provinces of Italy. This is important from a public health point of view, because it shows that, without analysing the mortality by district, we would have failed to identify differences that remain invisible when only larger geographical entities are taken into account.

A limitation of this study was that the municipal statistic offices only produce data for total mortality, without any indication of cause of death. This limits the analysis of the observed differences in mortality; however, the combination of the aspecific municipal data with the oldest, but cause-specific mortality data routinely compiled by the local health authorities may give enough information to hypothesise modifications of the risk factors active in a population. A second possible limitation was the difference in population size between districts that may, in a way, complicate the comparisons between districts. Similarly, when the population size of the district is low, the comparison of the differences in mortality with respect to the mortality of the town as a whole may be more difficult because of random variations. However, because the analyses we proposed in this paper were intended as a some kind of population health screening, this is only partially a serious limitation.

The differences in mortality rates we observed in Genoa may depend on various factors. Some studies linked health outcomes to pollution in Genoa [18,19,20,21,22], and one study found a relationship between socio-economic status and mortality by district [23]. Other studies identified a relation between socio-economic status and breast cancer survival [24] and between socio-economic status and cancer survival in the elderly [25]; extrapolating the results of these studies may indicate difficulties in access to health care for some categories of the Genoa population. 

Looking for differences in mortality rates between city districts is interesting from various points of view. The identification of such differences is important in the setting of environmental justice and social inequalities, because it may help provide the same protection from environmental and health hazards to the whole population and inform them about their health, while providing equal access to the decision-making process for health policy planning [26]. 

Differences in mortality rates between districts are linked to many urban phenomena; cities are composed of areas with considerable environmental and social heterogeneity, including serious poverty and very substandard living conditions that are associated with disease, mortality, and substandard levels of health. For example, the impact of the last financial crisis on mortality rates by socio-economic inequalities in small areas in Spain was analysed in a recent paper [27].

The local municipality and county authorities responsible for urban planning, local transport management, and social interventions may play an important role in enhancing population health and fighting environmental injustice and social inequalities. An example of the propositions a local administration can make is described in a paper published in 2020 [28] that presented the health impact assessment of an intervention to replace car trips with increased bicycle use and public transport in the greater Barcelona metropolitan area. The results showed that the intervention produced health benefits for travellers and the general city population, and could also help reduce greenhouse gas emissions. 

Municipal authorities are also responsible for green spaces; the link between green spaces and population health is evident from a review of 90 studies [29] and a paper about the association between green spaces, population health, and income deprivation [30]. 

It must be pointed out that mortality is also an indicator strongly linked to the development of a society; for example, a recent paper presented the association between worsening mortality rates and the rise of the Nazi Party in 1930s Germany [31].

The approach described in this paper is highly generalizable; we used data that are available at any municipal statistics office nearly everywhere in the world, the programs we developed in R to compute mortality rates by district are easily reproducible, and all computations can be performed in one month.

## 5. Conclusions

The use of all-cause mortality by municipal statistics offices to produce rapid district-related mortality rates may help local municipality and county administrators to analyse the health and social status of their populations as a whole and by district. In turn, a population’s awareness of the status of their health will make it possible for them to access the decision-making process for health policy planning.

## Figures and Tables

**Figure 1 ijerph-18-08250-f001:**
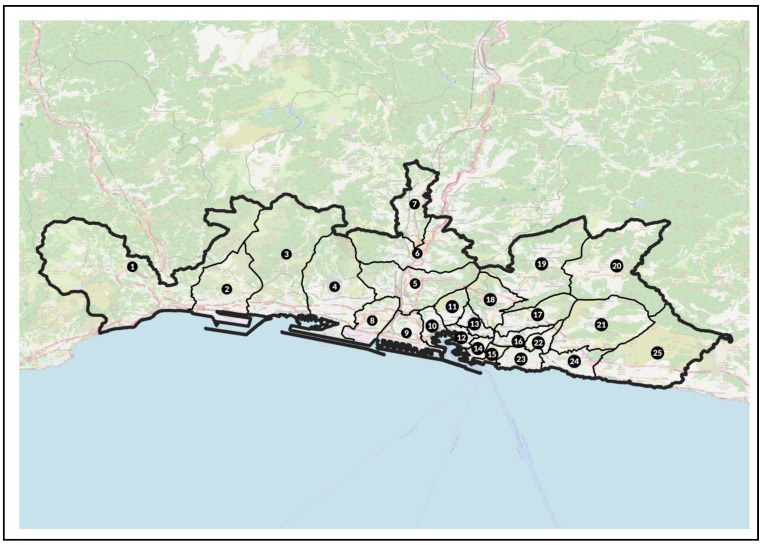
Genoa municipality and its 25 districts.

**Figure 2 ijerph-18-08250-f002:**
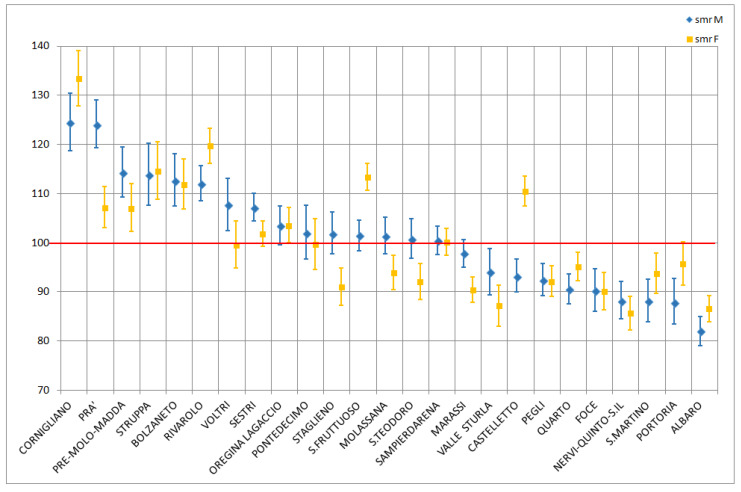
Standardised mortality ratios (SMRs) arranged by sex in the 25 districts of Genoa, 2009–2020, with 90% confidence intervals (CIs).

**Figure 3 ijerph-18-08250-f003:**
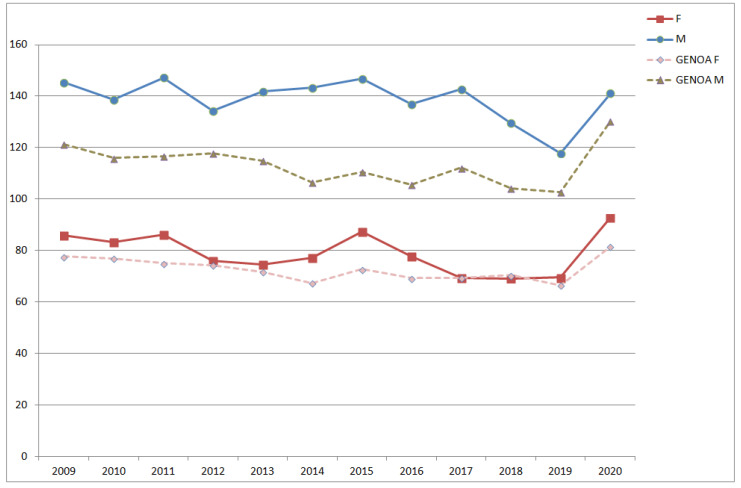
Annual standardised death rates (SDRs) in the district of Pra’ and the Genoa municipality. (F = females; M = males; GENOA F = females, whole town; GENOA M = males, whole town).

**Figure 4 ijerph-18-08250-f004:**
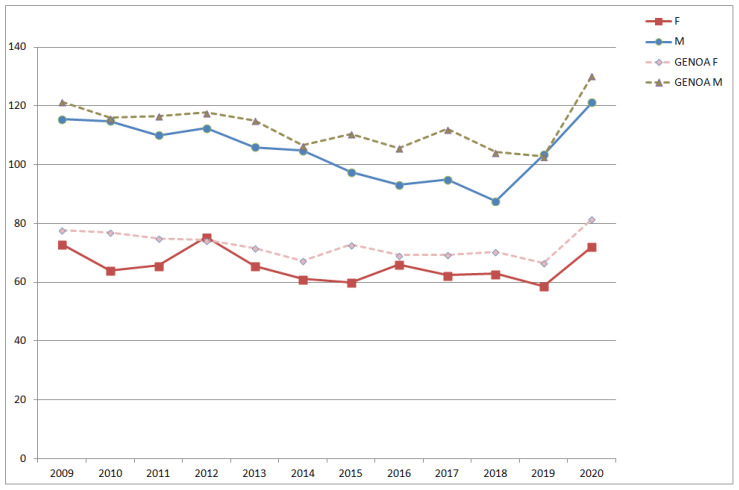
Annual SDRs in the district of Pegli and the Genoa municipality. (F = females; M = males; GENOA F = females, whole town; GENOA M = males, whole town).

**Figure 5 ijerph-18-08250-f005:**
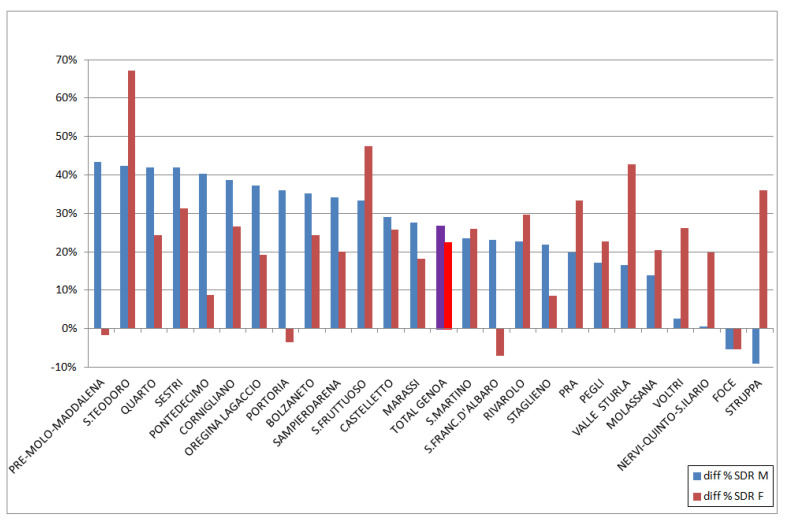
Percentage differences in SDRs among males and females, 2020 vs. 2019.

**Figure 6 ijerph-18-08250-f006:**
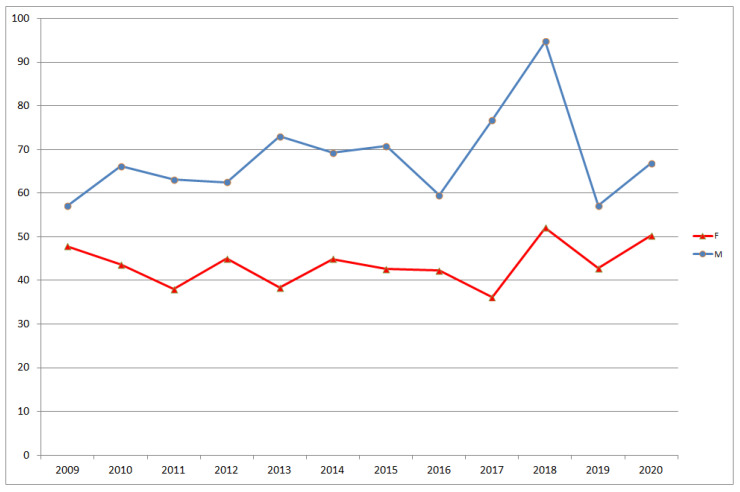
Temporal trend of differences between highest and lowest SDRs at district level, for males and females.

## Data Availability

This study used only aggregated data available on websites accessible to the public.

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
