# Peer review of "Municipality Data as a Rapid and Effective Tool to Analyse Spatial and Temporal Variations of All-Cause Mortality by Town District: The Experience in Genoa (Italy)"

_ijerph, 2021, doi:10.3390/ijerph18168250_

Round 1

Reviewer 1 Report

Omit 'early experiences' from the title (as this makes the work appear to refer to a historic analysis)

Please can you provide a map of Genoa and the districts contained within. Explain the terms of town, district and municipality. Imagine the reader doesn’t know your study area.

Be more consistent on the terminology and abbreviations. Refer to indirectly standardised mortality ratios (SMRs) for the within Genoa districts (due to smaller data input numbers) and standardised death rates (SDRs) for the larger area. Both of these methods control for age structure (in their different ways) and you have calculated separately for males and females.

SDRs are comparable over time. SMRs are not (though you could use the same age-specific rates for one time point and apply to a time series of death counts and age-structure). If you have sufficient data for the smaller areas such that SDRs are reliable then use these, if not, use SMRs.

As a general structure, I think it would come over well if you reported for the larger area first (and use SDRs) and then for the contained smaller areas (and use SMRs). Currently, it is unclear to me which geographies and measures are being reported and why.

When you have clearer information about the geography and measures, then I can have another look.

Author Response

See attached file for authors' reply.

Reviewer 2 Report

This manuscript set a primary objective of rapidly calculating standardized death rates by district, which was met; however, the discussion casts some doubt on whether the speed of the calculation (2-3 mo) is sufficient to meet needs based on the COVID-19 pandemic, when a major goal outlined in the introduction was response to infectious disease, based on historic use of similar calculations. 

Justification of the use of SMR to characterize the population, and selection of the 3 age groups used in the description, will be helpful. 

There are striking differences between some of the districts. The authors refer to health outcomes associated with pollution (17-21) and SES (22) by district, also for Genoa; it would be fair to discuss these in more detail in the context of the mortality results, which would give weight to the importance of the findings.

Given the discussion of timelines and mention that during infectious disease events like COVID-19, rapid timelines are even more important, a little more discussion on the feasible timelines for this type of analysis and their intersection with what would be needed to use the data to respond to specific threats would be helpful. 

A discussion of limitations and strengths of the study is important. 

Author Response

See attached file for authors' reply.

Round 2

Reviewer 1 Report

Thanks for the revisions regarding the study area (geographical terminology) and the methods used. Everything is much clearer now.